# The three-dimensional landscape of tumor-associated macrophages in reactive and neoplastic human lymph nodes

Aleksandar Vladisavljevic[1]*, Sonja Scharf[1,2,3], Hendrik Schäfer[1,2], Jörg Ackermann[3], Sylvia Hartmann[2], Ina Koch[3], Martin-Leo Hansmann[1,4], Patrick Wurzel[1]

**1** Institute of General Pharmacology and Toxicology, Goethe-University Frankfurt, Frankfurt, Germany, **2** Dr. Senckenberg Institute of Pathology, Goethe-University Frankfurt, Frankfurt, Germany, **3** Molecular Bioinformatics, Goethe-University Frankfurt, Frankfurt, Germany, **4** Frankfurt Institute for Advanced Studies, Frankfurt, Germany

\* al.vl@live.de

## Abstract

Macrophages play a crucial role in the homeostasis of lymph nodes. Their phenotypes and functions are dictated by the molecular composition of the microenvironment. The presence of tumor cells can influence the microenvironment, leading to changes in the cellular functions of tumor-associated macrophages (TAMs). Cellular alterations often correlate with histomorphometric and distributional changes. This study aims to characterize these pathomic modifications resulting from tumor cell infiltration by comparing them to reactive conditions. We assessed CD68[+] and CD163[+] TAMs in 160 three-dimensional (3D) images of human lymphoid tissue sections derived from 82 cases comprising six different diagnoses. To investigate TAM profiles, we employed a hybrid approach involving computer vision and graph theoretical algorithms. For calculating the histomorphometric features of TAMs, we utilized an image analysis pipeline with IMARIS Advanced Tracking Software. The distributions of TAMs were characterized using cell graphs. The incorporation of 3D tissue analysis in targeted areas of thick tissue sections revealed specific patterns of histomorphometric and distributional changes in TAMs. In chronic lymphocytic leukemia and diffuse large B-cell lymphoma, these distinctive alteration patterns demonstrated entity specificity. Furthermore, pathomic alterations displayed possible correlations with established functional aberrations in lymphomas. These findings imply that the pathomic properties of TAMs mirror their functional aberrations. Through validation and generalization via molecular pathological examinations, this approach has the potential to unveil and understand functional aberrations in histopathological assessments.

**Data availability statement:** We have provided the complete processed, image-level quantitative data used for all analyses as Supporting Information.

**Funding:** This work was supported by the Deutsche Forschungsgemeinschaft (HA6145/6-1 to SH; HA6145/7-1 to SH; HA6145/9-1 to SH; HA1284/18-1 to MLH). The funders had no role in study design, data collection and analysis, decision to publish, or preparation of the manuscript.

**Competing interests:** The authors have declared that no competing interests exist.

## Author summary

Macrophages play a pivotal role in the homeostasis of lymph nodes. The phenotype and function are intricately linked to the molecular environment. The emergence of tumors can disrupt this environment, inducing alterations in tumor-associated macrophages (TAMs). These changes are observable and analyzable through three-dimensional (3D) imaging. In this investigation, we analyzed 160 3D images of human lymph node tissue sections from six different diagnoses to elucidate the characteristics of TAMs. Employing a hybrid approach that integrates computer vision and graph algorithms, we aimed to identify entity-specific patterns.The findings revealed distinct TAM patterns in cases of chronic lymphocytic leukemia and diffuse large B-cell lymphoma. Known patterns of heightened interactions between TAMs and Hodgkin-Reed-Sternberg cells could be also correlated with revealed compact arrangements of TAMs in the microenvironment. This underscores that established functional aberrations of TAMs are, to some extent, mirrored in their pathomic properties. Overall, our results demonstrate that such hybrid workflows can provide new perspectives on TAM-related neoplastic aberrations and thereby contribute to a deeper understanding of tumorigenesis.

## 1. Introduction

Macrophages play a vital role in maintaining immune system homeostasis [1]. Beyond their physiological functions, they can also adopt tumor-promoting roles during tumorigenesis of various lymphomas [1,2]. Which lymphoma entities are affected by macrophage-driven pathology, how heterogeneous these alterations are, and whether they are mirrored in pathomic features remains unknown. Here, we address these gaps to advance understanding of the role of macrophages in lymphoma pathology.

Macrophages are a heterogeneous group of professional phagocytes which are integral to cellular homeostasis [1–3]. While ensuring the cellular and structural integrity of lymph nodes, macrophages participate in processes such as sensing, chemotaxis, phagocytosis, tissue repair, and adaptive stimulation [1]. To encompass their diverse functionalities, macrophages exist as distinct subpopulations, including classically activated macrophages (CAM/M1) and alternatively activated macrophages (AAM/M2) [1–5].

Within the immune system, M1 macrophages play a central role in driving pro-inflammatory responses. Through the secretion of pro-inflammatory cytokines, they induce strong inflammatory reactions and thereby contribute to the clearance of pathogens and tumor cells. In contrast, M2 macrophages exhibit anti-inflammatory properties. They secrete mediators that suppress inflammation, promote tissue remodeling, and are critically involved in wound healing, angiogenesis, and immune regulation [5,6,7].

Both macrophage subtypes arise from the maturation of monocytes. Their activation phenotype and functional properties are determined by the molecular composition of the microenvironment [1,6–9]. The presence of tumor cells can alter the molecular composition of the tumor microenvironment (TME) and thereby reprogram macrophages into so-called tumour-associated macrophages (TAMs) [2,4,9–17]. For example, tumors may promote the activation of M2 macrophages, creating an immunosuppressive environment that facilitates immune evasion [11,16]. Through functional modifications, TAMs can also actively contribute to the anastomosing process of tumor cells, promoting tumor angiogenesis, growth, and survival [11]. Additionally, TAMs can facilitate the invasion of tumor cells [10,11,16,18].

Although these potential functional adaptations of TAMs are known, the specific diagnoses in which they occur remain insufficiently defined. To date, diagnosis-specific investigations have largely focused on prognostic associations, with increased M2 macrophage density correlating with poorer patient outcomes [4,8]. However, functional differences are often associated with more subtle pathomic changes at the cellular level [12,13,15]. For example, predominantly phagocytic macrophages of the germinal center, so-called tingible-body macrophages, exhibit a significantly larger cell volume than T-zone macrophages [1,3]. Furthermore, based on known functional alterations, it is likely that alterations in the TME not only affect M2 activity but also cause broader shifts within the macrophage population. Building on these associations, the present study aims to go a step further.

First, we expand the population of macrophages under investigation. Conventional studies of M2 macrophage density typically rely on CD163 staining of M2 macrophages in human lymph node biopsies [4,8]. We supplement this approach with CD68 staining, a pan-macrophage marker that is also used as a proxy for M1 macrophages in some studies [8,7]. Second, the assessment of pathomic macrophage properties such as morphology in 2-µm thin sections is limited, since macrophages possess substantially larger cell volumes. To overcome this limitation, we apply three-dimensional (3D) confocal imaging of 20-µm thick sections, enabling accurate depiction of macrophage morphology and spatial distribution [17–20]. Here, macrophage morphology can be precisely evaluated using classical computer vision methods [17–20]. Building on this, their spatial distribution within the lymph node can be characterised through cell-graph approaches [21–23].

By integrating such advanced digital pathology with the novel 3D dataset, we establish a methodological framework for delineating diagnosis-specific pathomic alterations of macrophages, thereby enabling new functional hypotheses. This advances our understanding of the multifaceted role of macrophages in lymphoma pathology.

## 2. Results

To discern pathomic alterations in TAMs induced by a tumor, it is crucial to establish a baseline model encompassing histomorphometric and distributional traits of reactive TAMs. Consequently, we conducted an analysis and characterization of macrophages in non-neoplastic conditions, such as lymphadenitis (LA), differentiating between CD68+ macrophages and CD163+ macrophages. This allowed us to define morphological and distributional patterns within the B-zone, T-zone, and Medulla (Fig 1).

### 2.1. Macrophages under reactive conditions

Under reactive conditions, a distinct division into a B-zone and T-zone was consistently evident in all images, while the medulla was only partially distinguishable. Within the B-zone, a higher occurrence of CD68+ macrophages compared to CD163+ macrophages were observed, whereas the T-zone and medulla exhibited diffusely distributed CD68+ and CD163+ macrophages.

In the B-zone, CD68+ macrophages tended to adopt enlarged spherical morphologies when contrasted with CD163+ macrophages. Additionally, CD68+ macrophages in the B-zone maintained a greater distance and lower density in comparison to the T-zone or medulla (Fig 1). CD163+ macrophages were infrequently found within the B-zone.

Within the T-zone of LA, CD68+ macrophages formed smaller bodies and more spherical morphologies, maintaining a reduced macrophage-to-macrophage distance. Conversely, CD163+ macrophages in the T-zone exhibited elongated

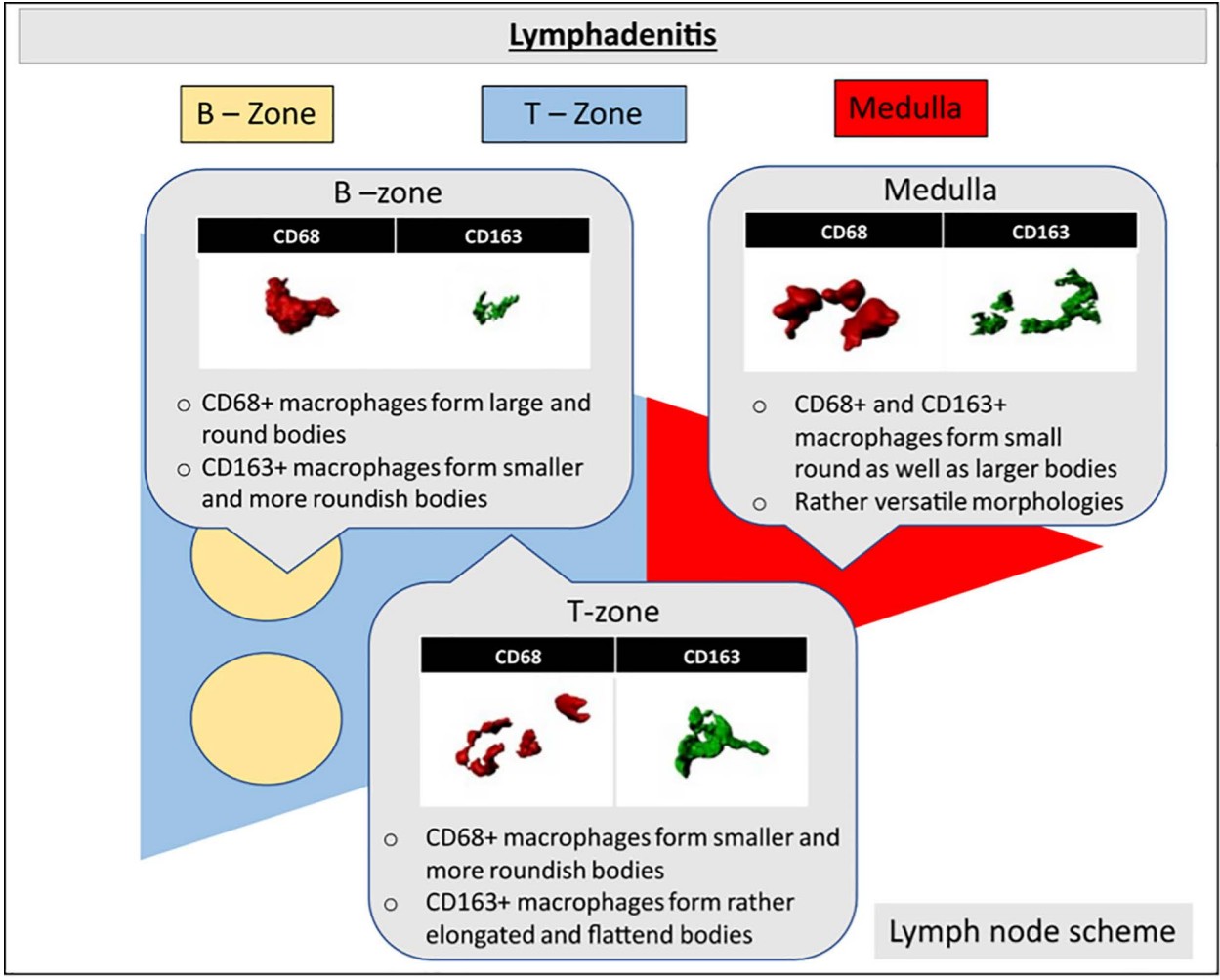

**Fig 1. TAM characteristics in reactive tissue.** Schematic representation of a human lymph node illustrating three-dimensional CD68$^+$ (red) and CD163$^+$ (green) macrophages in the B-zone (yellow), T-zone (blue), and medulla (red) in Lymphadenitis. CD68$^+$ macrophages exhibited large and round bodies in the B-zone of the lymph node, while displaying smaller and more versatile morphologies in the T-zone and Medulla of the lymph node in LA. Conversely, CD163$^+$ macrophages demonstrated primarily small and round bodies in the B-zone, with elongated and versatile morphologies observed in the T-zone and Medulla of the lymph node in LA.

and disc-shaped morphologies in contrast to their smaller and more spherical counterparts in the B-zone (Fig 1). In the medulla, both CD68$^+$ and CD163$^+$ macrophages displayed a versatile morphology, encompassing small round as well as large disc-shaped bodies (Fig 1).

## 2.2. Pathomic prototyping

This section examines pathomic alterations in TAMs across different lymphoma entities in comparison to reactive macrophages. To this end, TAMs were analysed in 3D images of reactive and neoplastic lymph nodes, as described in Section 4.3 ('Image Analysis and Graph Generation'). Morphological (surface-based) and spatial distribution parameters (cell graph-based) were extracted, comprising 24 features listed in Table 1. For each image, median values were calculated to generate a pathomic profile. Each feature was then tested for significant differences ($p < 0.05$)

**Table 1. Parameters of the pathomic profile.** The pathomic profile comprises the characterization of both the morphological attributes of a cell and its distribution characteristics. Morphological attributes were assessed separately for each staining. The source of each feature is provided in parentheses.

| No. | Feature Name | Feature Explanation | No. | Feature Name | Feature Explanation |
|---|---|---|---|---|---|
| Features describing Distribution | | | | | |
| 1 | CD68/CD163 Density | Density of cells inside of an image (Cell volume/ Image volume) | 4 | 3D packing via Estrada index | Compactness of the 3D arrangement of connected cells (NetworkX) |
| 2 | Average Connections | Average connections of cells inside the cell graph (NetworkX) | 5 | Isolated macrophages | Number of cells with zero connections (NetworkX) |
| 3 | Local Communication Efficiency | The average multiplicative inverse of the shortest path between two cells (NetworkX) | | | |
| Features describing CD68 and CD163 morphology | | | | | |
| 6 | Ellipticity (oblate) | Disc-shapedness of a cell (Imaris) | 11 | Sphericity | Roundness of cell (Imaris) |
| 7 | Ellipticity (prolate) | Cigar-shapedness of a cell (Imaris) | 12 | Volume | Volume of a cell in $\mu m^3$ (Imaris) |
| 8 | CD163/CD68 Intensity Mean DAPI | Mean intensity of a specific staining inside a cell (Imaris) | 13 | CD163/CD68 Antigen-Density | Density of expressed antigens on the surface of a cell (Sum of intensity/ surface) |
| 9 | CD163/CD68 Intensity Mean CD163 | Mean intensity of a specific staining inside a cell (Imaris) | 14 | Surface | Surface of a cell in $\mu m^2$ (Imaris) |
| 10 | CD163/CD68 Major Axis Length | Length of the longest axis of a cell in $\mu m$ (Imaris) | | | |

relative to reactive TAMs, thereby identifying entity-specific pathomic alterations (visualized as boxplots in S1 Fig). In Fig 2, each coloured cell denotes a significant alteration: red indicates an increase, green a decrease, and grey no significant change compared to reactive macrophages (S2 Fig provides a comprehensive overview of feature-wise p-values).

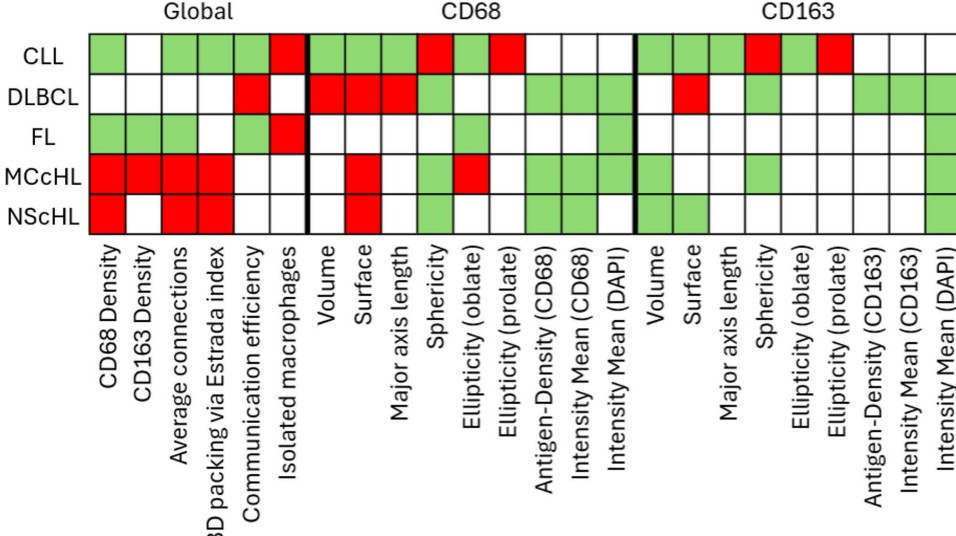

**Fig 2. Pathomic prototyping.** Scheme illustrating the pathomic prototype of each diagnostic entity based on the pathomic profiles. A green-colored box reflects a significantly reduced feature compared to reactive cases. Red boxes depict significantly increased features and grey-colored boxes highlight unaltered features.

Within the TME of HL, CD68+TAMs exhibited a significant increase. Conversely, CD163+macrophages displayed a significantly smaller volume in HL, while CD68+macrophages in MCcHL demonstrated an increased surface area with reduced CD68 antigen expression. Overall, TAMs in HL exhibited a highly connected and compact arrangement (Fig 2).

In CLL, CD68+TAMs within the TME were significantly reduced. The TAMs exhibited notably smaller and more spherical morphologies, with fewer contacts and a higher prevalence of isolated macrophages. Similar characteristics were observed for TAMs within the TME of FL. Unlike CLL, TAMs in FL primarily displayed regular morphologies and CD163+TAMs were significantly diminished (Fig 2).

TAMs in DLBCL displayed highly interconnected macrophages with a consistent density. CD68+TAMs presented significantly enlarged bodies with a lower density of CD68 antigens on the cell surface (Fig 2). It can be inferred that, across all entities except in CLL, there was, on average, a less intense DAPI staining of TAMs.

## 2.3. Pathomic diversity: Unique signatures and overlaps of TAM characteristics across diagnostic entities

To analyze pathomic diversity, we examined the classification patterns of a multiclass bagged decision tree classifier trained on the collected pathomic profiles. Whereas the previous analysis compared each neoplastic entity to reactive lymph nodes on a per-feature basis, the present analysis considers all features collectively across all diagnoses. This approach provides an intuition of which entities share pathomic commonalities, and which exhibit highly characteristic and unique pathomic alterations of TAMs.

The results are visualized in Fig 3 (left). The confusion matrix displays the percentage of cases classified into each diagnostic category, thereby indicating the classification accuracy. Each cell indicates how often a true class (rows) was predicted as a certain class (columns), relative to the total number of cases in that true class. Diagonal values indicate the

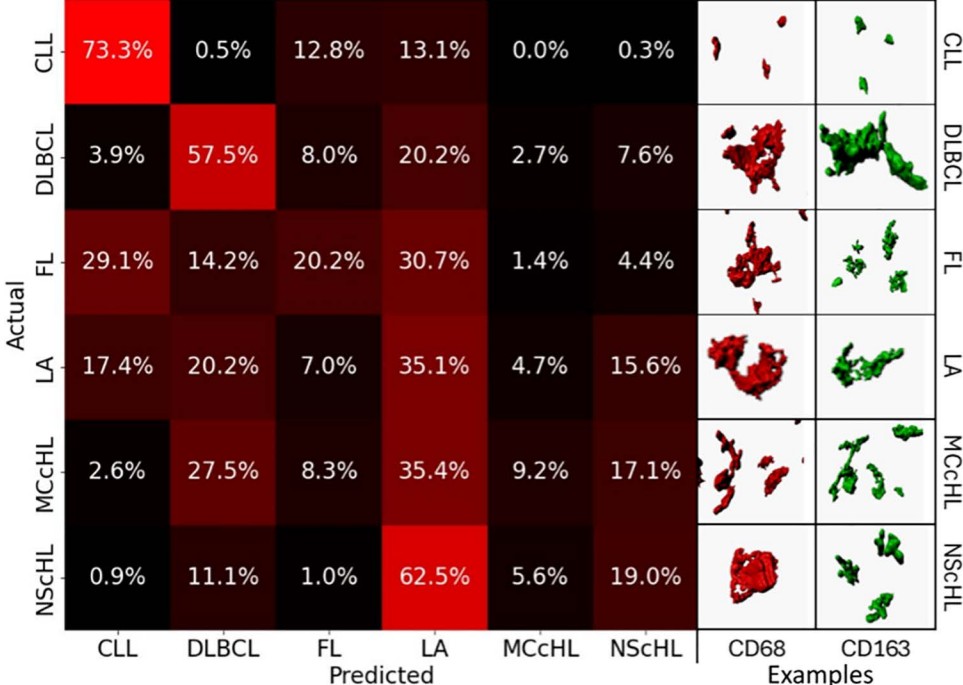

**Fig 3. Pathomic diversity.** The left-hand side displays the confusion matrix of the classification results. CLL and, up to a certain extent, DLBCL appear to exhibit specific alterations in macrophage morphology and distribution. The right-hand side showcases manually selected representative examples of CD68+ and CD163+ macrophages for each diagnosis.

proportion of correctly classified cases (accuracy), whereas off-diagonal values represent misclassifications, reflecting the degree of overlap and separation between diagnostic entities.

We observed unique patterns with notable accuracy for cases diagnosed as chronic lymphocytic leukemia (CLL) (73.5% accuracy) (Fig 3). Comparable results, though slightly less specific, were noted for diffuse large B-cell lymphoma (DLBCL) (57.5% accuracy). Cases identified as follicular lymphoma (FL) exhibited varied predictions, with cases rightly predicted (20.2% accuracy), cases predicted as CLL (29.1%), and cases predicted as LA (30.7%). Cases diagnosed as mixed cellularity classical Hodgkin lymphoma (MCcHL) were predominantly predicted as LA (35.4%), followed by DLBCL (27.5%) and nodular sclerosis classical Hodgkin lymphoma (NScHL) (15.6%). For NScHL, cases were predicted as LA (62.5%) and had a lower percentage of accurate predictions (19% accuracy), along with predictions as DLBCL (11.1%). Finally, in reactive cases diagnosed as LA, we observed a broad and diverse overlap with CLL (17.4%), DLBCL (20.2%), NScHL (15.6%), and accurately predicted cases (35.1% accuracy).

## 3. Discussion

In this study, we present the first three-dimensional pathomic characterization of tumor-associated macrophages (TAMs) in reactive and neoplastic human lymph nodes. By generating baseline profiles of reactive macrophages and comparing them to lymphoma subtypes, we identified diagnosis-specific morphological and spatial alterations. These results demonstrate that lymphoma biology leaves structural and organizational traces in macrophages and expand our knowledge beyond conventional density-based analyses and broad functional classifications.

Since macrophages in reactive lymph nodes already exhibit marked heterogeneity, we used this condition as a reference to identify pathomic properties relevant in the context of functional adaptation. Manual examination of CD68+ and CD163+macrophages also revealed localization as a strong source of variability (Fig 1). CD163+macrophages in the T-zone and medulla exhibited elongated morphologies suggestive of activation, consistent with previous studies [5,6]. CD68+TAM populations within the B-zone exhibited lower density and larger cell bodies compared to CD68+macrophages in the T-zone and medulla. CD68+TAMs in the B-zone resembled tingible body macrophages, predominantly displaying phagocytic properties [1,3]. In contrast, CD68+cells in the T zone displayed highly variable shapes, possibly due to the coexistence of M1- and M2-like macrophages [7,11].

Building on these qualitative distinctions, a pathomic profile was generated for each image/case using a hybrid approach involving morphological and distributional investigations (Fig 2). In the first step, we tested each feature of neoplastic TAMs for significant changes compared to reactive macrophages as part of pathomic prototyping.

In the HL, TAMs displayed increased density and a compact arrangement, consistent with the ability of Hodgkin-Reed-Sternberg cells to remodel their microenvironment through cytokines and direct cell-cell interactions [12,24]. In DLBCL, macrophage density did not increase, but CD68+cells were enlarged, indicating an enhanced phagocytic response to the high amount of debris caused by rapid tumor turnover [16,25–27]. In CLL, both the number and size of CD68+TAMs were reduced, consistent with the low apoptosis rate of CLL cells [14,28,29]. In FL, macrophage density was predominantly reduced, reflecting the similarly reduced apoptosis rate of follicular lymphoma tumor cells. [25,30]. However, cell morphology was not altered, suggesting less adaptation of TAMs in the context of FL. This may be due to the fact that, especially in low-grade FL, tumor cells remain predominantly confined to the germinal center, where the overall macrophage density is low [9,14]

These entity-specific observations were reflected in the analysis of pathomic diversity (Fig 3). Using a multiclass bagged decision tree classifier, we performed an all-vs-all classification and analyzed the accuracy per entity as well as the misclassification pattern [31–33]. CLL and DLBCL showed comparatively specific pathomic signatures, suggesting that activity-related changes are robustly captured by pathomic features. FL, in contrast, was frequently misclassified as CLL or LA, consistent with the reduced apoptosis in CLL and tumor confinement. [28,29]. HL predominantly showed a broad classification spectrum, with NScHL very frequently misclassified as LA. This was unexpected, as previous studies and

our own prototyping demonstrated specific HRS-TAM interactions [12,17]. The high variability of HL and the heterogeneity of reactive microenvironments could explain these results.

However, it's important to keep in mind that this analysis only examined a very small section of the neoplastic lymph node. This could, of course, also reveal the extent to which an incorrect selection or an insufficiently small section can influence the assessment of individual patients.

In summary, our results underscore the high heterogeneity of reactive macrophages and demonstrate that TAMs in lymphomas do not follow a uniform pathomic trajectory. CLL and DLBCL exhibit distinct signatures, whereas HL and FL strongly overlap with reactive states. Their alterations could therefore resemble physiological immune responses or reveal functional changes that may not be fully reflected in pathomic features. By providing the first diagnosis-specific 3D pathomic characterization of macrophages, our study lays a foundation for future investigations that link morphology and function and refine the understanding of the multifaceted role of macrophages in lymphoma biology.

## 4. Materials and methods

### 4.1. Dataset

Tissue samples were sourced from the former archive of the Reference and Consultation Centre of Lymph Node and Lymphoma Pathology at the Dr. Senckenberg Institute of Pathology in Frankfurt am Main. Human samples were obtained from the Ear-Nose-Throat Centre of the University Hospital Frankfurt am Main following routine tonsillectomy. All collected samples underwent anonymization, ensuring they could not be linked to any specific individual. The complete anonymized, processed image-level quantitative dataset underlying all analyses is provided in S1 Data. The utilization of these tissue samples was approved by the institutional guidelines of Goethe University Frankfurt. There are no legal or ethical objections. The ethics committee has issued a positive approval.

A total of 82 cases, encompassing 160 images, were examined, representing six diagnostic entities: LA (20 cases, 44 images), CLL (15 cases, 25 images), DLBCL (15 cases, 28 images), FL (11 cases, 20 images), MCcHL (9 cases, 16 images), and NScHL (12 cases, 27 images).

### 4.2. Sample preparation

The sample preparation followed the protocol outlined by Liebers et al. [17]. Formalin-fixed samples, embedded in paraffin, were sliced into thin sections measuring 18–33 $\mu m$. Subsequently, the paraffin sections underwent deparaffinization and hydration through xylenes and a graded alcohol series. The sections were then incubated for 30 minutes with VectaFluor Duet Reagent.

For the combined CD163 and CD68 immunostaining, we employed the Rabbit monoclonal antibody (Clone: EPR19518) (abcam, Cat. No. ab182422) against CD163$^+$ macrophages, and Monoclonal Mouse antibody (Clone KP1) Anti Human (Agilent, Cat. No. M0814) against CD68$^+$ macrophages. Cellular visualization was achieved using the VectaFluor Duett Double Staining labeling kit (Cat. No. DK-8818). Nucleic acid staining was conducted with DAPI (D9542 Sigma Aldrich, St. Louis, USA), while the Vector TrueVIEW Quenching Kit (Burlingame, CA 94010) was employed to minimize autofluorescence (Fig 5). All protocols were executed as per the manufacturer's recommendations, and negative controls were prepared for each sample.

The generation of 3D images was facilitated by a Leica SP8 Confocal microscope (Leica Microsystems, Wetzlar, Germany; Objective: HC PL APO 63x/1.3 GLYC CORR Cs2, Lasers: 405mm DMOD Compact, Green 488, Red 552). The chosen scan format was 1024 x 1024 pixels, with a Z-stack interval of 0.13 $\mu m$. The image size was 3250 x 3250 $\mu m$, and the pixel size was 0.167 $\mu m$ on both the X and Y axes (Figs 4 and 5).

### 4.3. Image analysis and graph generation

For image analysis, Imaris Advanced Tracking Software versions 9.2 and 9.5 (Bitplane AG, Zurich, Switzerland) were utilized. The "Create Surface" tool was employed to calculate the surfaces of CD163$^+$ and CD68$^+$ cells within 3D images.

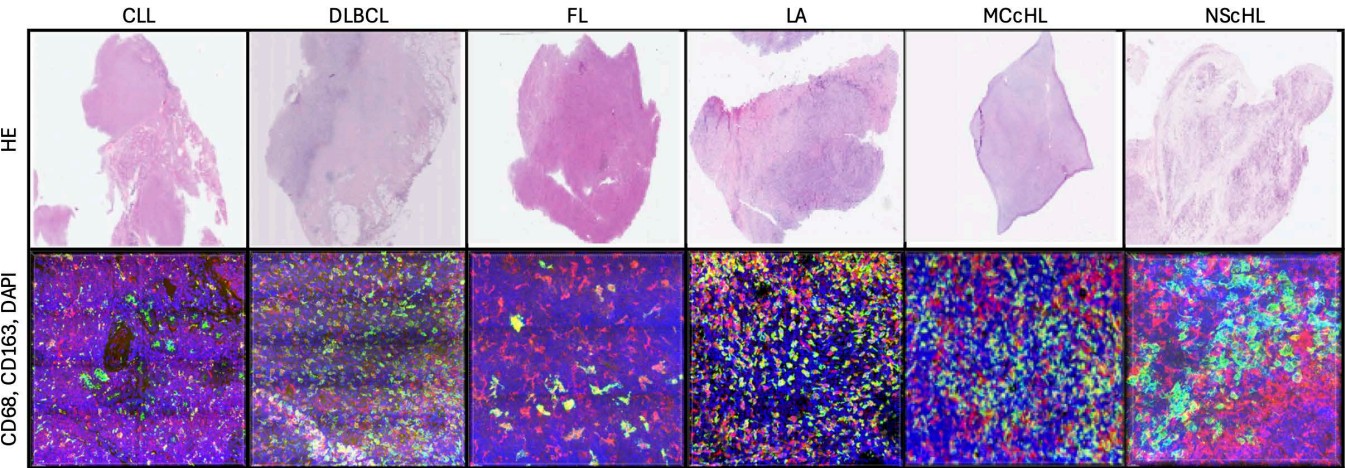

**Fig 4. Juxtaposition of two-dimensional microscopic pictures (HE) and the three-dimensional immunohistochemical reconstruction (CD68 – red, CD163 – green, DAPI – blue).**

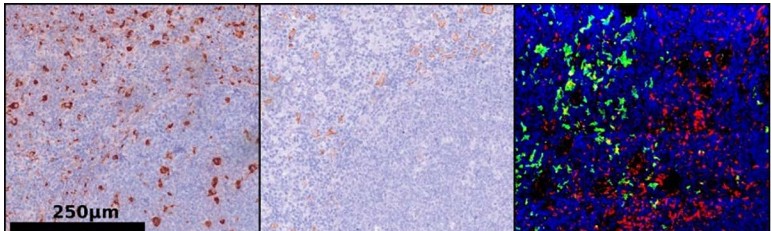

**Fig 5. Representative images of tissue samples.** Left – conventional 2D H&E-stained image; Middle – 2D image showing CD68+ and CD163+macrophages; Right – 3D visualization of CD68+macrophages (red), CD163+macrophages (green), and DAPI-stained nuclei (blue).

Each positively stained surface defined a volume object, referred to here after as a "volume". As part of the image analysis, morphological characteristics for each TAM were calculated and parameterized using Imaris. The cell density was calculated based on the summed cell volumes and the image size itself. Utilizing the previously localized CD163+ and CD68+ TAMs within individual images, we employed cell graphs to characterize the distribution of macrophages. We implemented a Python 3.7-based pipeline using NetworkX [34] to calculate cell graphs following the methodology outlined by Schaefer et al. [22]. Cell graphs were defined as geometric unit disk graphs. To establish edges, we applied an image-specific threshold based on the median major axis length of the identified TAMs. Based on the established cell graphs we calculated graph-based features with NetworkX to parametrize the distribution of TAMs [34]. Finally, the median of each single-cell-level feature was computed on a per-image basis. Combined with image-descriptive features (e.g., cell density), this resulted in a pathomic profile comprising 24 features per image (Fig 6 and Table 1).

### 4.4. Data science

Based on pathomic profiles, we evaluated neoplastic TAMs for significant pathomic alterations ($p < 0.05$) compared to reactive TAMs. Here, we compared each parameter of the pathomic profiles of each neoplastic entity separately with the pathomic profiles of reactive tissue samples using the Mann-Whitney U test. We also assessed whether pathomic features were increased or decreased in the median, allowing us to define pathomic prototypes for each neoplastic entity (Fig 2).

To evaluate pathomic diversity across all diagnostic entities, we evaluated the classification performance of a bagged decision tree classifier trained on pathomic profiles. To avoid overrepresentation, the dataset was downscaled to one image, i.e., the pathomic profile of the first section of a case. The final dataset was split into 75% training and 25% test images. To minimize the impact of dataset partitioning on classification performance, we performed 1000 iterations of random training-test splits [31–33] Figs 6, 7.

To evaluate classification performance, we generated receiver operating characteristic (ROC) curves for each diagnostic entity across 1000 training–test iterations. The averaged ROC curves demonstrated that the model performed above chance level across all classes, providing a reliable basis for assessing its discriminative ability (Fig 8A). To assess the interpretability of individual pathomic features, we computed SHAP (SHapley Additive exPlanations) values using a model-agnostic permutation explainer applied to a model trained without data splitting. This analysis showed that all types of pathomic features, both morphological and distributional, contributed to the classification, albeit to varying degrees (Fig 8B). These results indicate that the selected features are interpretable and relevant.

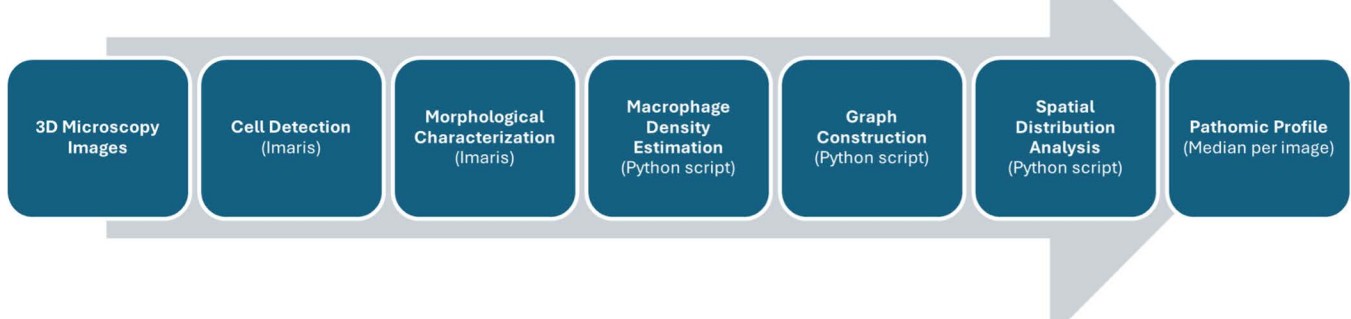

**Fig 6. Data extraction.** The figure illustrates the sequential steps from the 3D image data to the parameterized cell morphology and distribution per image, summarized in the pathomic profile. The pipeline is based on a hybrid approach combining computer vision-based cell detection using the Imaris software and self-implemented approach for graph-based description of cell distribution.

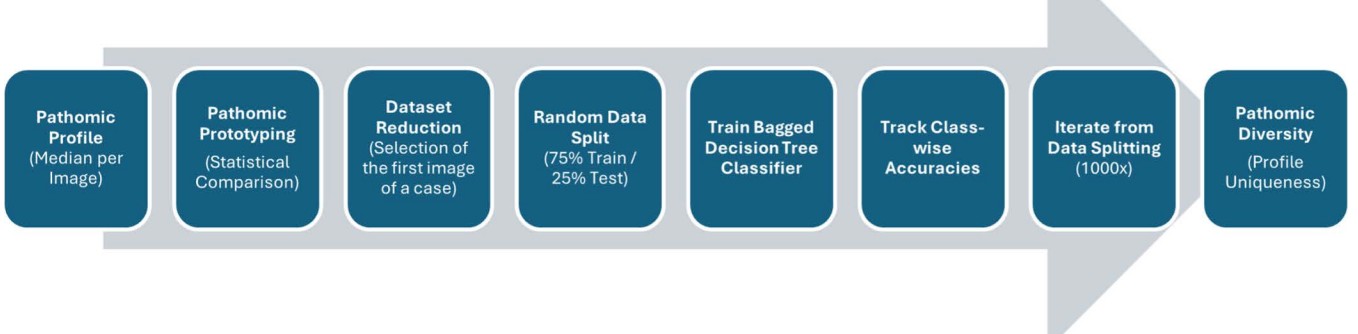

**Fig 7. Data analysis workflow.** The illustrated analysis workflow of pathomic profiles proceeds in two stages. First, diagnosis-centered statistical evaluation of pathomic features is performed, a process referred to as pathomic prototyping. This is supplemented by machine learning–based multiclass classification of the different diagnoses, termed here as pathomic diversity analysis.

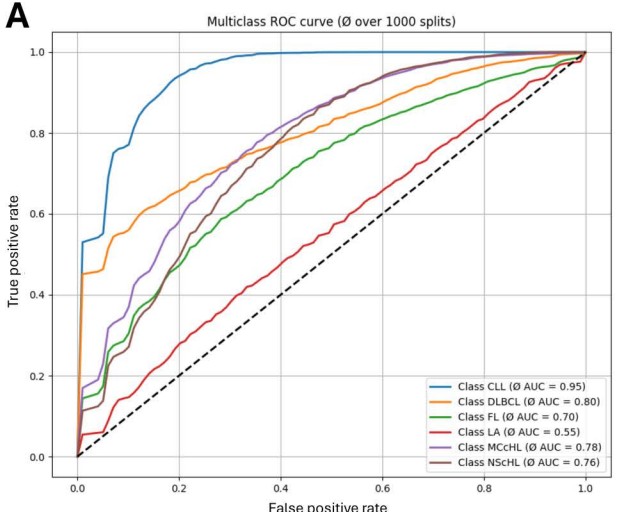
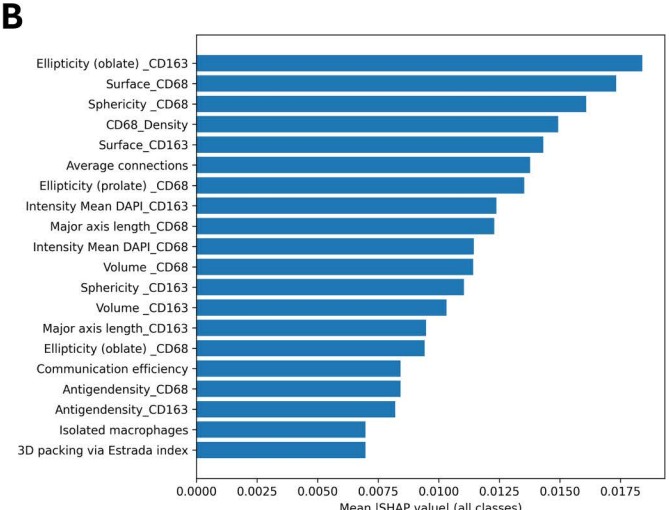

**Fig 8. Model evaluation. A:** One-vs-rest receiver operating characteristic (ROC) curves averaged across 1000 stratified train–test splits (75%/25%). Mean AUC values quantify the class-wise discriminative performance of the bagged decision tree classifier, with CLL showing the highest separability (AUC = 0.95) and LA the lowest (AUC = 0.55). **B:** Global feature-importance analysis based on mean absolute SHAP (SHapley Additive exPlanations) values computed from a model trained without data splitting. Higher SHAP values indicate stronger contributions of individual pathomic features to the predicted class probabilities. Both morphological and microarchitectural features contributed to the decisions made by the classifier.

To further evaluate the diagnosis-specific classification performance, we generated an adjusted confusion matrix by aggregating results from all 1000 runs. Each cell in the resulting confusion matrix represents how often a true class (rows) was predicted as a certain class (columns), relative to the total number of analyzed cases of the class. Correct classifications appear along the diagonal, while off-diagonal values indicate misclassifications, revealing the extent of overlap and differentiation between diagnostic entities. This approach allowed us to assess both the uniqueness of pathomic alterations for each diagnosis and the degree of shared pathomic features across different entities (Fig 3).

The full pipeline, incorporating image analysis, graph generation and data science, was implemented in Python 3.7, incorporating packages such as scikit-learn, SciPy, pandas, and NetworkX [32–35].

## Supporting information

**S1 Fig. Pathomic Prototypes of Diagnostic Entities Across Markers.** S1 Fig (a-c) - Boxplots depicting the pathomic prototype of each diagnostic entity based on pathomic profiles. (a) CD163 features, (b) CD68 features, and (c) global features Pathomic prototyping.
(PDF)

**S2 Fig. Scheme illustrating the pathomic prototype of each diagnostic entity based on the pathomic profiles, corresponding p-values.** All significant changes are highlighted in green. Statistical significance was defined as p < 0,05.
(PDF)

**S1 Data. Anonymized Macrophage Data per Image.** This table contains the complete processed, image-level quantitative data underlying all analyses reported in the manuscript.
(XLSX)

## Acknowledgments

Mr. Jörg Ackermann passed away before the submission of the final version of this manuscript. Mr. Aleksandar Vladisavljevic accepts responsibility for the integrity and validity of the data collected and analyzed. Special thanks to Yvonne Steiner, Susanne Hansen and Bianca Reisinger for the specimen preparation and imaging.

## Author contributions

**Conceptualization:** Martin-Leo Hansmann, Patrick Wurzel.

**Data curation:** Aleksandar Vladisavljevic, Sonja Scharf, Patrick Wurzel.

**Formal analysis:** Aleksandar Vladisavljevic, Patrick Wurzel.

**Funding acquisition:** Sylvia Hartmann.

**Investigation:** Aleksandar Vladisavljevic, Patrick Wurzel.

**Methodology:** Aleksandar Vladisavljevic, Sonja Scharf, Hendrik Schäfer, Jörg Ackermann, Ina Koch, Patrick Wurzel.

**Project administration:** Martin-Leo Hansmann.

**Resources:** Sylvia Hartmann, Martin-Leo Hansmann.

**Software:** Aleksandar Vladisavljevic, Patrick Wurzel.

**Supervision:** Martin-Leo Hansmann.

**Validation:** Aleksandar Vladisavljevic, Martin-Leo Hansmann.

**Visualization:** Aleksandar Vladisavljevic, Patrick Wurzel.

**Writing – original draft:** Aleksandar Vladisavljevic, Patrick Wurzel.

**Writing – review & editing:** Aleksandar Vladisavljevic, Sonja Scharf, Hendrik Schäfer, Jörg Ackermann, Sylvia Hartmann, Ina Koch, Martin-Leo Hansmann, Patrick Wurzel.

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
