## [Decision Letter · Decision Letter 0]

27 Jan 2025

Response to Reviewers
Revised Manuscript with Track Changes
Manuscript
**Journal Requirements:**

1. We ask that a manuscript source file is provided at Revision. Please upload your manuscript file as a .doc, .docx, .rtf or .tex.

**Additional Editor Comments (if provided):**
**Reviewers' Comments:**

Reviewer #1: The authors present pertinent findings regarding the morphology of TAMs in the context of malignancy pathologies and correlate them with the respective pathologies. While the research presents significant value, certain issues require attention before publication. Furthermore, the scope of PLOS Digital Health does not align with the proposed research. Consequently, I recommend the rejection of the manuscript. After revision, it may be submitted to the PLOS One Journal.

A notable deficiency in the manuscript is the absence of microscopy images. Although tri-dimensional reconstructions are presented, no “real” microscopy images are depicted, nor are 3D representations of such images. I propose that examples of each pathology be accompanied by reconstructions.

Additionally, a pipeline diagram would enhance the explainability of the process.

Regarding segmentations, while IMARIS Advanced Tracking Software is effective, the authors could enhance their workflow by utilizing Machine Learning Tools within the software. These tools offer reduced manual intervention and minimize the likelihood of errors. Additionally, object classification pipelines significantly enhance accuracy and improve the separation between classes.

The authors briefly mention the statistical analysis, including the number of samples per category and the corresponding p-values for each class. It would be beneficial to include this information in the text for clarity and completeness.

Comment on references:

References should be sequential. In line 53, there’s reference [1,29]. Why not [1,2]?

English suggestions and textual corrections:

Line 41: replace “scrutinized” by “analyzed”

Line 84: section is misnumbered

Reviewer #2: This is a very interesting topic and great addition to the digital health world. I appreciate that the authors combined both qualitative and quantitative analysis on the TAMs. However, I'd recommend that the analysis section be more detailed.

1. The objective of the study is to explore the relationship of macrophages and the tumors. This relationship isn't sufficiently explored. For example, Figure 2.2's numbers are pretty unimpressive, leading to doubts on whether the model is sufficiently constructed. Figure 3 included some features, but how did we arrive at that? Does it mean all those features were significant (whether positively or negatively)? It's unclear to me.

2. I'm not sure if the analysis was done using deep learning based on 3D images, or the dataset has been reduced from 3D images to measurements outlined table 1. It goes back to how to understand the result in Figure 2.2 (which shows that the classification isn't good at all, but I'm not sure if that is the focus of the paper)

3. It'll be good to have some literature research and outline what is unique about the study. Is it the data? Or the method? or the finding?

Reviewer #3: The identification of different types of macrophages in cancer is a crucial aspect, as each type is associated with distinct physiological and pathological phenomena. The apparent objective of this study was to provide a comprehensive overview of tumor-associated macrophages in reactive and neoplastic lymph nodes, which is undoubtedly an interesting and important area of research for understanding a significant type of human cancer.

However, the manuscript cannot be accepted for publication in PLOS Digital Health due to numerous deficiencies. It fails to meet the journal's minimum standards.

Introduction

The background provided in the manuscript is incomplete and lacks clarity. For example, the authors must clearly explain why it is important to study these two apparent types of tumor-associated macrophages. What roles do they play in tumors? What markers (e.g., CD68+ or CD163+) identify each type of macrophage? Unfortunately, this critical information is missing. The authors wrote as though the audience were pathologists already familiar with the importance of their work. While a comprehensive review of the field is not expected, the manuscript should at least explain the role of these markers, the functions of the macrophages in malignant tissues, and the significance of distinguishing between reactive and neoplastic lymph nodes. The introduction, as it stands, is insufficient, and this issue extends throughout the manuscript.

Methodology

Although the authors provide a detailed description of tissue preparation, the methodology related to the central objective of the study is poorly addressed. If the study focuses on the morphology of different macrophage types, the introduction should highlight any prior associations between macrophage morphology and pathological states (e.g., benign, malignant, or reactive). Additionally, questions such as whether reactive nodes are exclusively present in cancer and the importance of studying the distribution of tumor-associated macrophages within nodules remain unanswered. This contextual background is essential for interpreting Table 1.

The authors mention in section 4.4 (lines 236-237) that they studied 24 features “for each staining individually.” However, this statement is ambiguous. Did they analyze all the samples they had? How did they determine specificity? The manuscript does not clarify whether a specificity and sensitivity analysis was performed. It is well known that when studying pathological or immunohistochemical tissue samples, multiple images of the same tissue must be analyzed to ensure accuracy. A diagnosis or feature cannot be determined from a single image. This highlights a potential area for improvement using digital methods or AI in pathology.

The authors state in the results (line 121) and methods (lines 240-241) that they analyzed a single selected image per case 1,000 times. However, they do not explain how these images were selected, nor do they address potential biases in this selection process.

Results

The authors fail to adequately describe how they constructed the pathomic profile, which appears to be critical for interpreting their results. They vaguely mention a "hybrid approach" (lines 168-169) but do not explain how Figure 2 was created or how the specificity percentages shown in the figure were calculated. Similarly, they present a figure correlating the pathomic profile with pathological diagnoses but do not describe the process for constructing this profile.

In addition, they claim to have calculated specificity using the Mann-Whitney U test. Why didn’t they use ROC curves, which are standard for evaluating specificity and sensitivity? Perhaps they lacked quantitative data, but the manuscript does not provide enough detail to ascertain this.

Discussion and Conclusion

The discussion is weak, as it relies on flawed methodology and incomplete results. Furthermore, the manuscript lacks a clear conclusion. The authors’ statement in lines 184-187, suggesting that pathomic properties can be integrated into histopathological assessments to enhance functional descriptions of neoplastic aberrations, cannot be substantiated based on the presented data

Reviewer #4: Enhance Visualization: Provide more detailed and intuitive visual representations (e.g., 3D interactive models or simplified graphs) to support the findings.

Strengthen Discussion: Compare findings with similar studies in greater depth to highlight the novelty and broader implications of the work.

Supplementary Material: Add more detailed protocols or supplementary figures to aid reproducibility

**Figure resubmission:****Reproducibility:** To enhance the reproducibility of your results, we recommend that authors of applicable studies deposit laboratory protocols in protocols.io, where a protocol can be assigned its own identifier (DOI) such that it can be cited independently in the future. Additionally, PLOS ONE offers an option to publish peer-reviewed clinical study protocols. Read more information on sharing protocols at https://plos.org/protocols?utm_medium=editorial-email&utm_source=authorletters&utm_campaign=protocols

---

## [Decision Letter · Decision Letter 1]

9 Dec 2025

Response to Reviewers
Revised Manuscript with Track Changes
Manuscript

**Journal Requirements:**

**Additional Editor Comments:**

1. I recommend including ROC-based evaluation, such as one-vs-rest ROC curves for each diagnostic entity, to demonstrate the classifier’s performance.

2. A feature-importance analysis-e.g., shapley value or SHAP values, would help clarify which pathomic features most strongly influence the bagged decision tree predictions.

**Reviewers' Comments:**

**Comments to the Author**

Reviewer #4: All comments have been addressed

Reviewer #5: All comments have been addressed

publication criteria?

Reviewer #4: Yes

Reviewer #5: Yes

3. Has the statistical analysis been performed appropriately and rigorously?

Reviewer #4: Yes

Reviewer #5: Yes

4. Have the authors made all data underlying the findings in their manuscript fully available (please refer to the Data Availability Statement at the start of the manuscript PDF file)?

Reviewer #4: Yes

Reviewer #5: Yes

5. Is the manuscript presented in an intelligible fashion and written in standard English?

Reviewer #4: Yes

Reviewer #5: Yes

Reviewer #4: (No Response)

Reviewer #5: There have been a number of valid points raised by the reviewers. The paper covers a number of interesting areas. I think that the authors have made a good attempt to address the points raised.

**Do you want your identity to be public for this peer review?** For information about this choice, including consent withdrawal, please see our Privacy Policy

Reviewer #4: No

Reviewer #5: No

**Figure resubmission:**

**Reproducibility:**To enhance the reproducibility of your results, we recommend that authors of applicable studies deposit laboratory protocols in protocols.io, where a protocol can be assigned its own identifier (DOI) such that it can be cited independently in the future. Additionally, PLOS ONE offers an option to publish peer-reviewed clinical study protocols. Read more information on sharing protocols at https://plos.org/protocols?utm_medium=editorial-email&utm_source=authorletters&utm_campaign=protocols

---

## [Editor Report · Decision Letter 2]

24 Jan 2026

The three-dimensional landscape of tumor-associated macrophages in reactive and neoplastic human lymph nodes

PDIG-D-24-00348R2

Dear Mr. Vladisavljevic,

We are pleased to inform you that your manuscript 'The three-dimensional landscape of tumor-associated macrophages in reactive and neoplastic human lymph nodes' has been provisionally accepted for publication in PLOS Digital Health.

Best regards,

Iqram Hussain, Ph.D.

Academic Editor

PLOS Digital Health